# Efficacy of a Novel Treatment Approach for Obstructive Sleep Apnea

**DOI:** 10.3390/biomedicines13102413

**Published:** 2025-10-02

**Authors:** Brandon Hedgecock, Max Kerr, Jenny Tran, Ben Sutter, Phillip Neal, Gilles Besnainou, Erin Mosca, Len Liptak

**Affiliations:** 1Sleep Better Austin Dental Sleep Medicine Clinic, Austin, TX 78613, USA; bhedgecock@sleepbetteraustin.com (B.H.); mkerr@sleepbetteraustin.com (M.K.); jenny.tran.dmd@gmail.com (J.T.);; 2Carl R. Darnall Army Medical Center, Killeen, TX 76544, USA; 3Lariboisiere Hospital, 75010 Paris, France; gillesbesnainou@gmail.com; 4ProSomnus Sleep Technologies, Pleasanton, CA 94588, USA

**Keywords:** obstructive sleep apnea, oral appliance therapy, mandibular advancement device, mandibular repositioning device, mandibular advancement splint, sleep disordered breathing, sleep medicine, snoring, respiratory medicine, precision medicine, oximetry, personalized medicine, sleep medicine

## Abstract

**Objective**: This study evaluates the efficacy of a novel approach to oral appliance therapy (“OAT”) for the treatment of obstructive sleep apnea (“OSA”). This novel approach utilizes a systemized, oximetry-informed, treatment protocol and a precision-custom oral appliance. **Methods**: Sixty consecutive patients diagnosed with OSA were treated at Sleep Better Austin (“SBA”) using a structured, multi-step protocol and a precision-custom oral appliance (ProSomnus EVO). Baseline and post-treatment apnea–hypopnea index (“AHI”) values were compared using a matched-pair design. The primary outcome was the percentage of patients achieving a residual AHI of <10 events/h. Secondary outcomes included severity classification improvement. **Results**: In total, 90% of patients achieved the primary endpoint, and 87% improved by at least one severity classification. The mean AHI improved by 63% from baseline with the precision-custom OAT in situ (*p* < 0.001). In the moderate-to-severe subgroup, AHI improved by 70%, with 100% of severe patients achieving a residual AHI of <20 and a ≥50% improvement, without patient preselection. No serious adverse events were reported, and all patients continued therapy at follow-up. **Conclusions**: Precision-custom OAT, when delivered through a standardized clinical protocol informed by oximetry, can be a highly effective and well-tolerated treatment for OSA. These findings support its broader adoption as a non-invasive alternative to continuous positive airway pressure (“CPAP”) and surgical interventions, particularly for patients seeking personalized, high-compliance solutions.

## 1. Introduction

Obstructive sleep apnea is a highly prevalent and underdiagnosed disease affecting an estimated one billion people worldwide including 176 million in China, 84 million in the United States, 52 million in India, 49 million in Brazil, 26 million in Germany, and 24 million in France with an apnea–hypopnea index (“AHI”) greater than five events per hour [1,2]. Left untreated, OSA contributes to adverse health outcomes, a diminished quality of life, and economic burden [2,3,4].

The pathophysiology of OSA is well-established, involving the recurrent collapse of the velopharyngeal and oropharyngeal airway segments during sleep, resulting in intermittent hypoxia and sleep fragmentation [5,6]. Research establishes that 97% of airway collapse events involve the velopharynx segment while 56% of events involve the oropharynx segment [6].

OSA severity is commonly characterized using the apnea–hypopnea index (“AHI”), which guides therapeutic decisions and outcome assessments [7]. The AHI, which is the average count of apnea and hypopnea events per hour, is used to quantify the severity of the disease. An AHI between five to fifteen events per hour is considered mild sleep apnea; fifteen to thirty events per hour is considered moderate; and more than thirty events per hour is considered severe [7]. Success is commonly determined by how well therapy improves the AHI relative to the baseline.

Despite the availability of treatment modalities such as continuous positive airway pressure (“CPAP”), surgeries and surgically implanted devices, and traditional oral appliance therapy (“OAT”), each approach presents limitations. CPAP, while effective, is often poorly tolerated [8,9]. Surgical interventions, though potentially beneficial, are invasive, costly, and frequently require preselection procedures such as drug-induced sleep endoscopy [10,11]. Traditional OAT offers a non-invasive alternative with relatively better patient acceptance, yet its broader adoption is hindered by inconsistent efficacy and a lack of standardized clinical data [7].

This study addresses these gaps by evaluating a novel and systemized approach to OAT that integrates a precision-custom oral appliance with a structured, oximetry-informed treatment protocol. Unlike prior investigations that often rely on semi-custom devices or variable clinical workflows, this research introduces a repeatable, multi-step protocol designed to optimize mandibular positioning and therapeutic outcomes. The precision-custom oral appliance used in this study is fabricated from individualized digital records and avoids prefabricated components, thereby reducing variability and enhancing personalization. Oximetry studies inform device titration decisions, as opposed to relying on subjective assessments of therapeutic response.

Importantly, this investigation is among the first to demonstrate the efficacy of precision-custom OAT across all severities of OSA—including moderate and severe cases—without patient preselection. The matched-pair study design, use of objective oximetry data, and rigorous follow-up process contribute to the robustness of the findings. By achieving high rates of therapeutic success and classification improvement, this research provides compelling evidence for the broader adoption of precision-custom OAT as a safe, effective, and scalable alternative to CPAP and surgical interventions.

## 2. Materials and Methods

This study was a retrospective chart review of patients from a single dental sleep medicine practice with multiple locations. The design is organized and described according to the Population, Intervention, Comparison, Outcomes (“PICO”) format [12]. For clarity, the Intervention component is divided into two sections. The first describes the treatment protocol. The second describes the medical device used in this study.

### 2.1. Population

Sixty patients diagnosed with OSA were referred to the Sleep Better Austin (“SBA”) organization for OAT. All sixty patients completed the treatment protocol. This study population was segmented by OSA severity classification to facilitate post hoc analyses. Forty-four of the sixty patients, 73.3% of the study population, presented with mild OSA. Sixteen patients, 27% of the study population, presented with moderate or severe OSA, with 18.3% and 8.3% being moderate and severe, respectively.

### 2.2. Intervention, Protocol

Upon OSA diagnosis by their managing physicians, patients were referred to SBA for OAT. The protocol described in Figure 1 was implemented consistently for each patient enrolled in this study, across seven SBA clinic locations in the greater Austin, Texas metropolitan area (Southwest, Central, Cedar Park, Georgetown, Kyle, Killeen, and Waco), involving multiple SBA team members.

The first step of the SBA treatment process was the new patient consultation. The new patient consultation featured an examination for contraindications, a review of medical insurance coverage, and the taking of digital records of each new patient’s oral structures. Digital records of oral structures included digital scans of each patient’s upper and lower teeth, the relevant gingival anatomy, and a digital scan for each patient’s mandibular position that is associated with maintaining velopharyngeal and oropharyngeal dilation. The therapeutic mandibular position was established at 50–60% of each patient’s maximum protrusive range, using a George Gauge tool (Great Lakes Dental Technologies, Tonawanda, NY, USA). These digital records were electronically transmitted to the oral device manufacturer (ProSomnus Sleep Technologies, Pleasanton, CA, USA) for fabrication of the precision-custom OAT device.

Delivery of the precision-custom OAT device was the second step of the SBA process. The delivery appointment was scheduled for approximately two weeks following the initial consultation. During this appointment, patients were instructed on how to insert, remove, and care for their OAT device, and how to perform compliance exercises.

The third step was the follow-up appointment. The initial follow-up appointment was conducted approximately 2 weeks after delivery of the OAT device. Regular follow up appointments were scheduled monthly, to review instructions, compliance, symptoms, and results of the oximetry studies (NightOwl, Resmed, San Diego, CA, USA). Oximetry study results were utilized to objectively assess response to treatment and optimize OAT device titration. Patients were referred to their managing physician for confirmation testing. Patients were placed on an annual recall schedule if the managing physician confirmed successful response to treatment.

### 2.3. Intervention, Oral Appliance

A FDA cleared, precision-custom OA (ProSomnus EVO, Pleasanton, CA, USA), as depicted in Figure 2, was used to treat patients enrolled in this study.

Precision-custom means that each precision-custom device is 100% custom made from the oral records for each individual patient [13]. Unlike traditional oral appliances that rely on prefabricated, non-custom components, such as screws, mechanical hinges, nylon rods, or elastomeric straps, to reposition, stabilize, and titrate the mandible, precision-custom devices do not have prefabricated items [14]. Avoiding prefabricated items, and the modification steps required to embed them into devices, eliminates tolerance stacks, optimizing personalization and reducing variability in performance [15].

This OA is also entirely made from a material that satisfies the US Pharmacopeia’s requirements for the medical grade class VI designation [16]. Class VI represents the US Pharmacopeia’s highest standard for medical device material biocompatibility.

This precision-custom OA also uses a familiar and clinically proven dual-post design for repositioning, stabilizing, and titrating the mandible into the prescribed therapeutic location. Titration is accomplished with a stepwise sequence of trays, each with a different, prescribed, protrusive setting. This stepwise titration approach offers many benefits to both the provider and the patient including definitive titration, easier communication, enhanced durability, smaller volume in the mouth, and the elimination of metal components.

### 2.4. Comparison

This investigation had a matched-pair study design. It compared baseline OSA values for each patient against residual OSA values with the therapeutic oral device in situ.

Baseline OSA, assessed using the AHI (events per hour), was the primary basis for comparison.

### 2.5. Outcomes

Providers critically evaluated success by comparing the actual treatment outcomes relative to pre-determined performance goals. Performance goals were established by referring physicians, and according to normal and customary criteria, specifically as follows:Primary Performance Goal:
○A total of 65% of patients with a residual AHI of <10 events per hour
Secondary Performance Goals:
○% of patients that improved by at least one OSA severity classification○% of patients with a residual AHI of <10 and a >50% reduction from baseline○% of severe patients with a residual AHI < 20 and a >50% reduction from baseline


The primary performance goal of 65% of patients achieving the primary endpoint of an AHI of <10 events per hour with the OA in situ was based on the literature [7].

Compliance and safety were evaluated during follow-up visits, based upon oral examinations and subjective patient feedback.

### 2.6. Statistical Analysis

Statistical analyses were conducted to evaluate the efficacy of the precision-custom oral appliance therapy (OAT) in reducing the apnea–hypopnea index (AHI) among patients with obstructive sleep apnea (OSA). Given the matched-pair study design, where each patient served as their own control (baseline vs. post-treatment) and the non-normal distribution of AHI values confirmed through exploratory data analysis, non-parametric statistical methods were selected to ensure robust and valid inference.

The Wilcoxon signed-rank test was used to compare baseline and post-treatment AHI values. This test is appropriate for paired, non-normally distributed data and evaluates whether the median difference between paired observations is significantly different from zero. It was applied to both the overall study population and the moderate-to-severe OSA subgroup to assess treatment efficacy.

Descriptive statistics—including mean, median, standard deviation, and interquartile range (IQR)—were calculated to summarize the central tendency and variability of AHI values before and after treatment. These metrics provide a comprehensive view of the data distribution and support interpretation of the treatment effect.

All statistical tests were two-sided, with a significance level set at α = 0.05. Analyses were performed using GraphPad Prism v10.0 (GraphPad Software, LLC, Boston, MA, USA), a validated statistical software platform commonly used in biomedical research.

The selection of non-parametric methods and matched-pair analysis was guided by the study’s retrospective design, the nature of the outcome variable (AHI), and the goal of minimizing assumptions about data distribution. This approach ensures that the reported treatment effects are statistically sound and clinically meaningful.

## 3. Results

Figure 3 shows a box-and-whisker plot comparing the baseline AHI values (before treatment) with the residual AHI values (after treatment with precision-custom OAT). At baseline, the median AHI was 10.3 events/h. The mean AHI was 14.0 ± 11.3 events/h with an IQR of 9.7 events/h. The range was 5.0 to 69.8 events/h. The distribution shows a right-skewed pattern, with a long tail toward the higher AHI values, indicating variability in the baseline severity among patients.

The box-and-whisker plot in Figure 3 also describes the post-treatment residual AHI distribution. Median AHI is 3.0 events/h. Mean AHI is 4.2 ± 3.8 events/h with an IQR of 3.9 events/h. The range was 0.0 to 18.9 events/h.

Post-treatment values are significantly lower, with the mean and median values falling below the clinical threshold of 10 events/h. The distribution is tighter and symmetric, indicating reduced variability and a consistent therapeutic effect across the cohort.

The Wilcoxon signed-rank test was used to compare the baseline and residual AHI values. The result was *p* < 0.001, indicating a statistically significant reduction in the AHI with the precision-custom OAT.

Treatment with the precision-custom OAT improved the mean AHI by 63% relative to baseline. Further, with the precision-custom OAT, IQR was compressed, and whisker length was reduced relative to baseline values.

The mean baseline AHI for the mild cohort was 8.8 ± 2.7 (range: 5.0–14.9) events per hour. The mean baseline AHI for the moderate-to-severe cohort was 28.2 ± 13.6 (range: 15.1–69.8) events per hour.

In total, 54 of 60 (90%) patients achieved the primary threshold for therapeutic success, defined as an AHI of <10. This result of 0.9 [95% CI 0.79–0.96] met the performance target of 0.65 (65%) of patients achieving a residual AHI of <10. Figure 4 displays the matched-pair change in the AHI.

Figure 4 illustrates individual patient responses to precision-custom OAT, according to the AHI metric. Fifty-eight of sixty (97%) patients demonstrated an improvement in AHI with precision-custom OAT. The two patients who did not experience a decrease in their AHI had mild OSA at baseline and a negligibly higher AHI (still in the mild range) with OAT. Seventy-three percent of the total study population, 44 of 60 patients, achieved the secondary performance goal of a residual AHI of <10 and a ≥50% improvement over baseline. Five of five (100%) severe patients improved below Scher’s criteria threshold, with a residual AHI of <20 and a >50% reduction.

For the 27% of patients diagnosed with moderate or severe OSA, the mean AHI improved by 70%, 21.1 events per hour, from a baseline of 28.2 ± 13.6 (median 23.3 [IQR 15.4]) to 7.1 ± 5.3 (median 7.1 [IQR 8.6]) with OAT. A Wilcoxon signed-rank test indicated a significant reduction in the AHI from baseline to residual, with OAT in situ (*p* < 0.0001). Additionally, all matched-pair differences were positive, indicating that the residual values were consistently lower than the baseline values. The maximum residual AHI for the moderate-to-severe subgroup was 18.9 events per hour. The minimum residual AHI was 1.6 events per hour.

In total, 12 of 16 (75%) patients with baseline moderate-to-severe OSA achieved a residual AHI of <10 with OAT. Twelve of sixteen (75%) patients with baseline moderate-to-severe OSA achieved a residual AHI of <20 with a ≥50% reduction from baseline, and all five (100%) patients classified with baseline severe OSA achieved a residual AHI of <20 with a ≥50% reduction from baseline.

Figure 5 is a Sankey flow diagram, which visualizes the changes in OSA severity classifications for each individual patient—from baseline to residual—with precision-custom OAT.

At baseline, 73% of patients were classified with mild OSA, 18% with moderate OSA, and 8% with severe OSA. The distribution reflects a cohort predominantly comprising mild OSA patients, with a smaller but clinically significant subset of patients classified with moderate and severe OSA.

With treatment, a substantial (73%) number of patients shifted to the normal classification defined as an AHI of <5 events/h. A total of 15% of patients were classified into the mild OSA category with treatment. A substantial number of patients who were classified with moderate or severe OSA transitioned into the mild category. Only one patient remained in the moderate OSA category. And zero patients were classified with severe OSA with treatment.

In total, 85% of patients improved by at least one severity classification and 94% (15 of 16) of patients classified with severe or moderate OSA at baseline improved by at least one category. All 100% of patients with severe OSA improved by below the severe classification with treatment. The mean improvement in classification with treatment was 1.7 strata, indicating substantial therapeutic impact.

After follow-up appointments, 60 of 60 patients continued therapy. No patients required unscheduled interventions for adverse side effects. Any side effects reported, including dry mouth, tooth soreness, jaw soreness, or general discomfort, were non-serious, transient, and did not compromise the continuation of treatment.

## 4. Discussion

The findings of this study demonstrate the clinical significance and originality of a novel, precision-custom OAT protocol for the treatment of OSA. Delivered through a structured, oximetry-informed protocol, this approach resulted in a mean AHI reduction of 63%, with 90% of patients achieving the primary therapeutic goal of a residual AHI of <10. This outcome exceeded the predefined performance target of 65%, with a lower 95% confidence interval of 79%, underscoring the robustness and reliability of the results.

Importantly, this study contributes new evidence to the field by evaluating a fully systemized treatment model that integrates precision-custom oral appliances with objective oximetry data for titration. Unlike prior investigations that often rely on non-custom or semi-custom devices and subjective, patient-reported assessments, this research utilized a repeatable, multi-step clinical protocol and a precision-custom oral device fabricated from individualized digital records, eliminating prefabricated components and reducing variability in performance [13,14,15,16,17]. The use of medical grade class VI materials [12,16] and a dual-post titration design further enhanced device performance, likely contributing to high patient compliance and minimal side effects.

The results are particularly compelling in the moderate-to-severe OSA subgroup, where the mean AHI improved by 70%, and 100% of severe patients achieved a residual AHI of <20 with a ≥50% reduction from baseline. These outcomes challenge the conventional assumption that oral appliances are less effective in severe cases and suggest that precision-custom OAT, when delivered through a structured protocol, may be a viable alternative to CPAP and surgical interventions for a broader patient population. This aligns with prior studies reporting the high efficacy of precision-custom OAT devices [13,16,17], and contrasts with investigations involving other oral devices that demonstrated lower levels of efficacy [18,19,20].

The relatively high compliance rate and absence of serious adverse events further highlight the tolerability and practicality of this approach. Compared with CPAP, which is often associated with side effects and low adherence [8,9], and surgical options that carry risks and costs [9,10], precision-custom OAT offers a compelling balance of efficacy, safety, and patient-centered care. These findings are consistent with a previously reported head-to-head crossover study comparing the mean disease and mean risk alleviation of precision-custom oral devices and CPAP as frontline treatment [21].

This study also emphasizes the importance of protocol standardization in enhancing treatment efficacy. The SBA model—featuring detailed consultations, digital oral records, and regular follow-ups with home sleep testing—demonstrates how structured care delivery can optimize therapeutic outcomes and reduce variability. This contrasts with traditional OAT, where inconsistent workflows and limited clinical data have hindered performance and adoption [7].

Despite these strengths, the study has limitations. The sample size of sixty patients, while sufficient for demonstrating statistical significance, may not fully represent the broader OSA population, particularly given the predominance of mild OSA cases (73.3%). Additionally, the study was conducted at a single organization (SBA), which may introduce site-specific biases related to provider expertise or patient demographics. The reliance on home sleep testing for assessing treatment response, while practical, may not be as precise as polysomnography, the gold standard for AHI measurement. This study utilized single-night studies, which has been associated with misclassification of OSA severity relative to multi-night testing [22,23]. Furthermore, research suggests that the AHI alone may not be a reliable predictor of health outcomes in OSA patients [24]. Several comparative studies conclude that OAT is non-inferior to CPAP for blood pressure [7,25] and other important outcome measures [26], and that OAT is highly efficacious at improving sleep apnea specific hypoxic burden below the 60% min/h threshold that is associated with a normalized, versus elevated, level of residual health risk from OSA [21,27].

Future research should address these limitations by including larger, more representative populations and incorporating polysomnography, multi-night testing, and/or more predictive biomarkers such as hypoxic burden, for more robust outcome validation. Longitudinal studies are needed to evaluate the durability of AHI reductions and compliance over time. Additionally, comparative studies pitting precision-custom OAT against CPAP or surgical options in randomized controlled trials could further clarify the relative efficacy and cost-effectiveness of these therapies. Exploring the impact of precision-custom OAT on quality of life, comorbidities, and economic outcomes would also strengthen the case for broader adoption of this approach.

In total, this study presents compelling evidence for the efficacy, safety, and scalability of a novel, precision-custom OAT protocol. By demonstrating significant improvements across all severity levels, high compliance, and minimal side effects, it advances the field and supports the broader adoption of personalized, non-invasive therapies for OSA.

## 5. Conclusions

This study provides compelling evidence that a novel, precision-custom oral appliance therapy (OAT), when delivered through a structured, oximetry-informed clinical protocol, can be a highly effective, safe, and well-tolerated treatment for obstructive sleep apnea (OSA). The matched-pair design and use of objective outcome measures revealed a statistically and clinically significant reduction in the AHI across all severity levels, with 90% of patients achieving a residual AHI of <10 and 85% improving by at least one severity classification.

From a clinical perspective, these findings are particularly impactful. The protocol’s success in treating moderate and severe OSA—without patient preselection—demonstrates that precision-custom OAT may be a viable alternative to CPAP and surgical interventions, especially for patients seeking personalized, non-invasive solutions. The high compliance rate and absence of serious adverse events further support its potential for broad clinical adoption.

This research is original in its integration of a fully systemized treatment workflow, individualized device fabrication, and objective oximetry-guided titration. By eliminating prefabricated components and standardizing clinical delivery, the approach reduces variability and enhances therapeutic precision. These innovations address longstanding limitations in traditional OAT and contribute meaningful advancements to the field of sleep medicine.

In summary, this study advances the understanding of how precision-custom OAT can be optimized for real-world clinical practice. It highlights the importance of personalization, protocol consistency, and data-driven titration in achieving superior outcomes. These findings lay the groundwork for future research and support the broader implementation of precision-custom OAT as a frontline therapy for OSA.

## Figures and Tables

**Figure 1 biomedicines-13-02413-f001:**
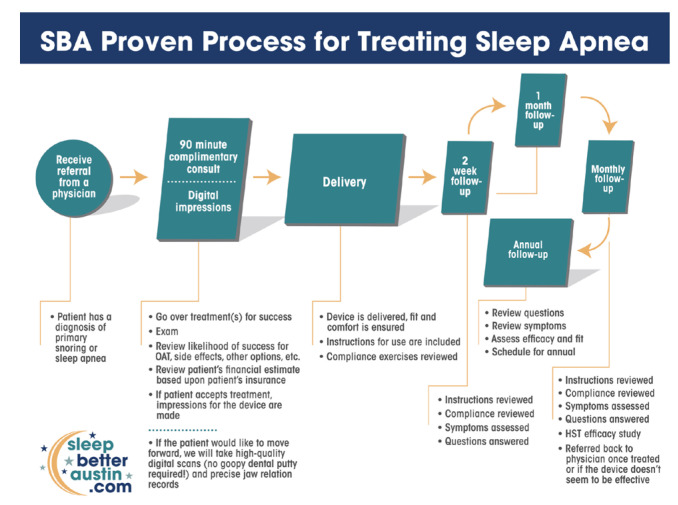
The Sleep Better Austin treatment protocol map.

**Figure 2 biomedicines-13-02413-f002:**
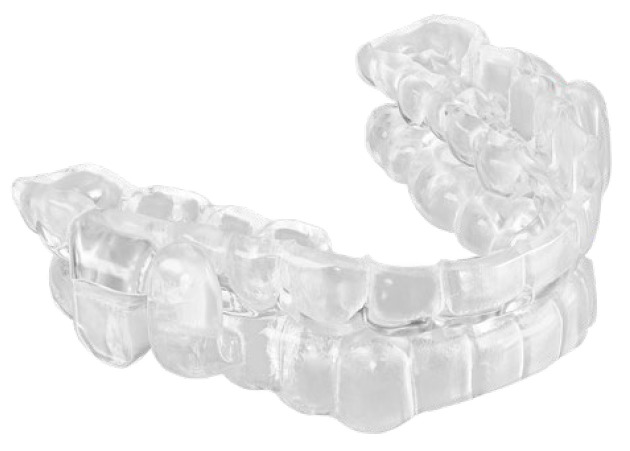
ProSomnus EVO Precision-custom Oral Appliance.

**Figure 3 biomedicines-13-02413-f003:**
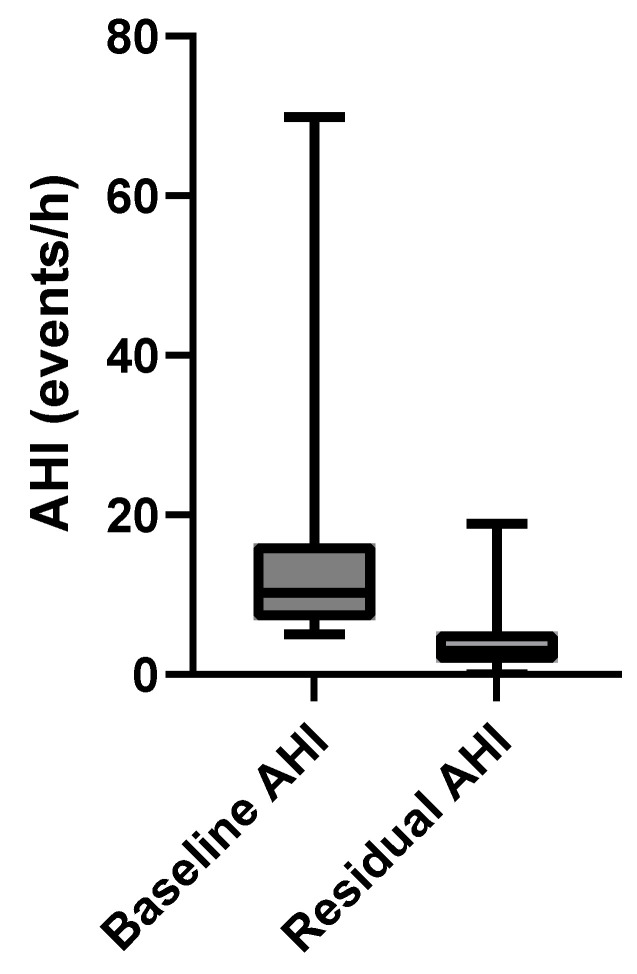
Box-and-whisker plot showing baseline and residual AHI.

**Figure 4 biomedicines-13-02413-f004:**
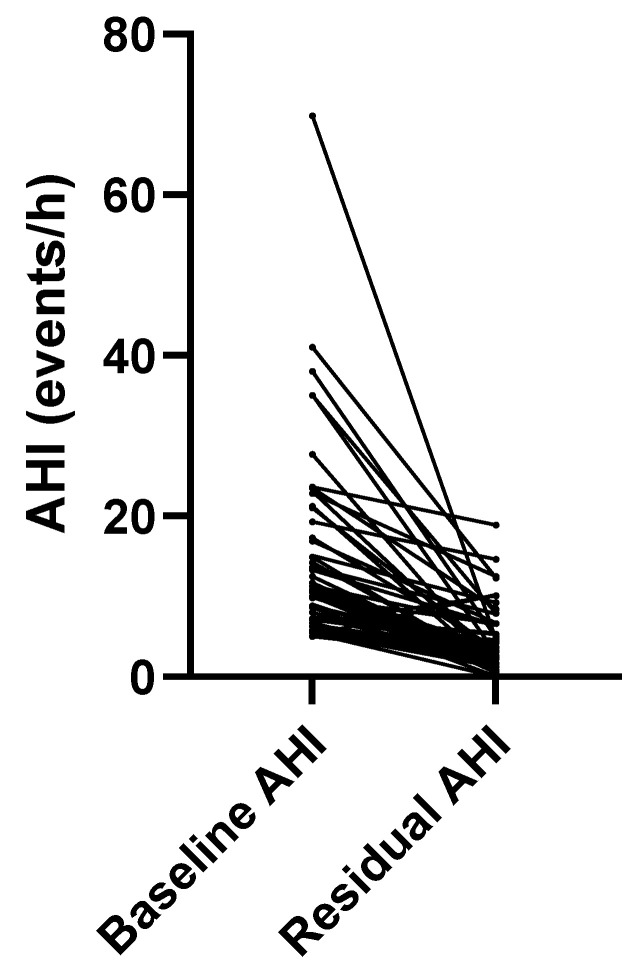
Individual patient responses to OAT.

**Figure 5 biomedicines-13-02413-f005:**
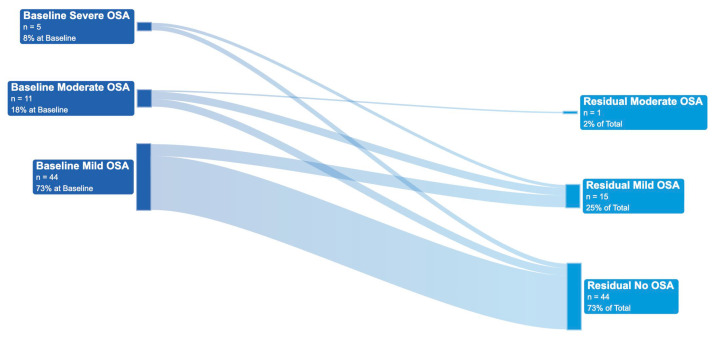
Sankey flow diagram of change in OSA severity classification with OAT.

## Data Availability

The data presented in this study are available from the corresponding author upon reasonable request.

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
