# Peer review of "Efficacy of a Novel Treatment Approach for Obstructive Sleep Apnea"

_biomedicines, 2025, doi:10.3390/biomedicines13102413_

Round 1
Reviewer 1 Report
Comments and Suggestions for Authors
Efficacy of a Novel Treatment Approach for Obstructive Sleep Apnea - please provide an aim and describe the necessity to perform this study.
Materials and Methods provide an Ethical Committee number.
Intervention, Protocol is too long. Please shorten it and provide essential facts.
Figure 1. ProSomnus EVO Precision-custom Oral Appliance is original or provide the source.
Lines 117-126 are unnecessary.
The outcomes should be clarified and more focused on the research.
The statistical analysis should be better described - what tests were performed and why.
Figure 1. Box-and-whisker plot showing baseline and residual AHI should be explained and discussed in the Discussion section.
Chart 3. Change in OSA Severity Classification, by Patient can be presented as a supplementary table.
Discussion: What is the originality?
The conclusion should be more clinically oriented and clearer.
The references are too few and many are outdated.
Reviewer 2 Report
Comments and Suggestions for Authors
Dear authors,
Thank you so much for submitting your paper to the prestigious journal Biomedicines.
I like the topic of the paper but some changes has to be implemented prior being accepted.
Please find bellow my suggestions and remarks that hopefully will be useful for you, in order to increase the quality of the manuscript.
- Lines 4-5 – academic degrees should not be included after the names of the authors. You may include them in the affiliation. For a better visibility please include ORCID.
- Lines 6-9 – Please be more descriptive regarding the affiliation. It’s not clear what kind of institution are.
- Please include the e-mail of the corresponding author. The paper doesn’t include a corresponding author.
- Line 14 – You are using the abbreviation OSA but you didn’t use the full name before in the text. Please include the abbreviation at Line 12.
- Line 62 – Please include the ethical approval number and send to the assistant editor a copy of it.
- Line 77 – That’s a figure a not a chart. Please revise.
- At lines 116 and 165 you state that both are „Figure 1” which cannot be. Please pay more attention and revise accordingly.
- Line 153 - Please revise the title of the section into „Results”
- Line 179 -That is not a chart but a table. Please revise it according to the technical requirements of the journal. Make sure it is cited in the text.
- Line 202- Discussion section should be elaborated as there is a lot of literature on this topic. 23 references are insufficient.
- Please rephrase the entire Conclusion section in such a way that will reflect your personal findings and their impact on optimizing current clinical protocols.
- Line 279 – please revise the authors contribution accordingly to the journal and IJCME authorship criteria.
- Line 293 – Please mention the ethical committee approval number.
- Line 298 – There is no corresponding author.
- Please include a table of all abbreviations as well, in order to increase the impact of the paper among readers.
- Please revise all references in the MDPI style.
- Please make sure that all Figures and Tables are properly cited in the text.
Positive Aspects:
1. The relative novelty of the topic;
2. Potential clinical impact of the findings;
Major Issues:
1. Lines 58-59 – Please indicate the multiple locations.
2. Please elaborate the material and methods section
3. Device description lacks sufficient detail to understand what “precision-custom” means and how it may differ from existing devices. What is different? What is original?
4. Potential bias — possible conflicts of interest or funding sources should be declared, especially given the device-specific focus.
5. Statistical reporting is incomplete. Please refine the statistics.
6. Line 202- Discussion section should be elaborated as there is a lot of literature on this topic. 23 references are insufficient for this topic.
7. Please rephrase the entire Conclusion section in such a way that will reflect your personal findings and their impact on optimizing current clinical protocols.
This manuscript has potential but, from my point of view, was written in a superficial manner.
Please revise the manuscript accordingly. In this form is not suitable for publication.
Reviewer 3 Report
Comments and Suggestions for Authors
This study was a retrospective chart review of patients from a single dental sleep medicine practice with multiple locations in Austin, Texas (USA). The study design was organized and described according to the Population, Intervention, Comparison, Outcomes (“PICO”) format. 60 patients diagnosed with OSA were referred to the Sleep Better Austin (“SBA”) organization for oral appliance therapy (OAT). 45 patients (73.3%) presented with mild OSA (AHI 5-15). 11 patients (18.3%) with moderate (AHI 15-30) and 5 patients (8.3%) with severe OSA (AHI>30).
The first step of the SBA OAT treatment process was a 90 minutes consultation. The consultation included an oral/dental examination to ascertain suitability for OAT, and an evaluation for contraindications and risks of side effects. The second step of the SBA process was delivery of the precision-custom OAT device. The third step of the SBA process was the initial follow up appointment that was conducted approximately 2 weeks after delivery of the OAT device. The fourth step of the SBA process was the monthly follow up sequence. Patients were scheduled for monthly follow up visits to review instructions, compliance, and symptoms. A type-3 home sleep test (NightOwl, Resmed, San Diego, CA) was utilized to assess response to treatment and optimize titration.
An FDA cleared, precision-custom OA (ProSomnus EVO, Pleasanton, CA) was used to treat patients enrolled in this study. Precision-custom means that each device is 100% custom made from the oral records for each individual patient.
The study examines an established treatment approach by which patients diagnosed with sleep apnea can be provided with a mandibular advancement device (OAT). The results are presented in a much too extensive and complicated manner, especially in Chart 3. Even if no new results on the effectiveness of mandibular advancement devices in sleep apnea therapy are presented here, the treatment process and the use of a particularly high-quality mandibular advancement device are presented, which ensures reliable and high-quality care for the patient. The article is also helpful because it includes and highlights processes that influence sustainability and therapy adherence.
Round 2
Reviewer 1 Report
Comments and Suggestions for Authors
The paper has been improved; the reviewer's indications have been followed.
Reviewer 2 Report
Comments and Suggestions for Authors
The paper was improved accordingly.